# Impact of Proton Pump Inhibitor Use on Progression-Free and Overall Survival in Cancer Patients Undergoing Immune Checkpoint Inhibitor Therapy: A Systematic Review and Meta-Analysis of Recent Studies

**DOI:** 10.3390/cancers17132228

**Published:** 2025-07-03

**Authors:** Giuliana Ciappina, Alessandro Ottaiano, Mariachiara Santorsola, Emanuela Esposito, Fabiola De Luca, Carlotta Giorgi, Concetta Zito, Anna Paola Capra, Patrizia Carroccio, Nicola Maurea, Vincenzo Quagliariello, Irene Campo, Maria Ilenia Passalacqua, Dalila Incognito, Irene Cacciola, Pierluigi Consolo, Massimiliano Berretta

**Affiliations:** 1Department of Medical Sciences, Section of Experimental Medicine, University of Ferrara, 44121 Ferrara, Italy; giuliana.ciappina@unife.it; 2Division of Innovative Therapies for Abdominal Metastases, Istituto Nazionale Tumori IRCCS Fondazione G. Pascale, 80131 Naples, Italy; a.ottaiano@istitutotumori.na.it (A.O.); mariachiara.santorsola@istitutotumori.na.it (M.S.); 3Department of Chemical, Biological, Pharmaceutical and Environmental Sciences, University of Messina, 98166 Messina, Italy; emanuela.esposito@unime.it (E.E.); fabiola.deluca@unime.it (F.D.L.); annapaola.capra@unime.it (A.P.C.); 4Laboratory for Technologies of Advanced Therapies (LTTA), Department of Morphology, Surgery and Experimental Medicine, Section of Pathology, Oncology and Experimental Biology, University of Ferrara, 48033 Ferrara, Italy; carlotta.giorgi@unife.it; 5Cardiology Unit, Department of Clinical and Experimental Medicine, University of Messina, 98166 Messina, Italy; concetta.zito@unime.it; 6School of Specialization in Medical Oncology, Department of Human Pathology “G. Barresi”, University of Messina, 98125 Messina, Italy; patrizia.carroccio@studenti.unime.it (P.C.); irene.campo@studenti.unime.it (I.C.); dalila.incognito@studenti.unime.it (D.I.); 7Division of Cardiology, Istituto Nazionale Tumori-IRCSS-Fondazione G. Pascale, 80131 Naples, Italy; n.maurea@istitutotumori.na.it (N.M.); vincenzo.quagliariello@istitutotumori.na.it (V.Q.); 8Division of Medical Oncology, AOU “G.Martino” Hospital, University of Messina, 98166 Messina, Italy; mariailenia.passalacqua@polime.it; 9Department of Clinical and Experimental Medicine, University of Messina, 98166 Messina, Italy; irene.cacciola@unime.it (I.C.); pierluigi.consolo@unime.it (P.C.)

**Keywords:** immunotherapy, immune checkpoint inhibitors, proton-pump inhibitors, PPIs, solid tumors, microbiota

## Abstract

Proton pump inhibitors (PPIs) are widely prescribed medications that may interfere with the efficacy of immune checkpoint inhibitors (ICIs) in patients with solid tumors. This systematic review and meta-analysis evaluated the impact of concomitant PPI use on survival outcomes in patients receiving immunotherapy. The results suggest a potential detrimental effect of PPIs on ICI efficacy, highlighting the need for critical assessment of supportive medications during cancer immunotherapy. Among the potential underlying mechanisms, growing evidence suggests that the gut microbiota, which can be altered by PPIs, plays a plausible role in immune modulation and therapeutic response. Further prospective studies are needed to clarify causality and guide clinical practice.

## 1. Introduction

The gut microbiota consists of over a trillion microorganisms, which form an ecosystem responsible for the production of metabolites involved in maintaining the host’s homeostasis [1]. Especially in solid tumors of the gastrointestinal tract, the activity of the microbiota can be involved in some processes implicated in tumorigenesis, such as inflammation, DNA damage, and immune invasion [2]. Several studies have also shown that the health status of the gut microbiota is correlated with the response to anticancer treatments, particularly with immunotherapy [3,4,5,6]. Immunotherapy has radically changed the survival rates of many solid tumors, particularly those historically characterized by poor prognosis, such as melanoma, non-small cell lung cancer (NSCLC), and urothelial tumors. Although immunotherapy was initially directed primarily toward patients with advanced-stage disease, there is a growing trend toward its earlier implementation, given the substantial efficacy demonstrated by immune checkpoint inhibitors (ICIs) even in earlier stages of disease, such as in NSCLC [7]. It is well known that tumors are capable of evading the immune system’s activity through the use of checkpoints, which are physiological pathways present in the body to regulate autoimmune processes. ICIs are drugs that can block this immune evasion mechanism, reactivating the immune system and its response to cancer [8]. Due to the increasing relevance of immunotherapy, attention has been focused on drugs that may interact with the activity of ICIs, particularly in patients who exhibit poor response rates. Among the most studied drugs are corticosteroids, whose use during immunotherapy has a detrimental effect, resulting in lower survival rates [9,10]. Given the role of the gut microbiota in regulating the immune system’s activity, it has been hypothesized that drugs that alter the state of eubiosis in the body may also be responsible for poor responses to ICIs [11]. Among these drugs, those most extensively studied are antibiotics and proton pump inhibitors (PPIs). PPIs are among the most commonly prescribed drugs worldwide. Their use is similarly high in patients with solid tumors, predominantly to prevent potential adverse effects induced by medications that increase the risk of gastric irritation or bleeding, collectively known as bleeding risk–increasing drugs (BRIDs), such as nonsteroidal anti-inflammatory drugs (NSAIDs) [12]. PPIs were associated with decreased progression-free survival (PFS) and overall survival (OS) in NSCLC and melanoma patients receiving ICIs [13,14], although the impact of these drugs is still debated. This meta-analysis was conducted to evaluate the impact of PPIs on survival outcomes in patients undergoing immunotherapy, regardless of the cancer site.

## 2. Materials and Methods

This study presents a systematic review and meta-analysis aimed at evaluating the association between PPIs use and time-to-event outcomes in patients receiving ICIs. Following the 2020 Preferred Reporting Items for Systematic reviews and Meta-Analyses (PRISMA) recommendations [15], the review was conducted according to a pre-defined and methodologically sound protocol, which was prospectively registered in the PROSPERO database (hosted by the National Institute for Health Research, London, UK) under registration number CRD420251027476. The completed PRISMA 2020 checklist is available as Appendix A.

### 2.1. Search Strategy and Selection Criteria

For this meta-analysis on the role of concomitant use of PPIs and immunotherapy in patients with cancer, we searched relevant publications in PubMed, Scopus, and the EMBASE library database. The search keywords was constructed as follows: [“immunotherapy” OR “immune checkpoint inhibitors” OR “PD-L1 antibody” OR “PD-1 antibody” OR “CTLA-4 inhibitor” OR “pembrolizumab” OR “atezolizumab” OR “ipilimumab” OR “nivolumab” OR “durvalumab”] AND [“proton pump inhibitor” OR “omeprazole” OR “pantoprazole” OR “esomeprazole” OR “lansoprazole” OR “concomitant medication”] AND [“survival” OR “overall survival” OR “progression free survival”]. The search was conducted from November 2022 to January 2025. A manual search strategy was employed, capitalizing on the strengths of human-guided review to enhance the precision and contextual relevance of study selection. Six investigators were assigned to two independent teams to carry out the process. The following items were the inclusion criteria for this meta-analysis: (1) retrospective or prospective clinical studies; (2) studies focusing on the prognostic value of PPIs administration in cancer patients treated with anti-PD-1/PD-L1/CTLA-4 monotherapy or in combination; (3) available data for PFS or OS with a hazard ratio (HR) and 95% confidence interval (95% CI). The following items were the exclusion criteria for this meta-analysis: (1) animal experiments; (2) reviews, comments, letters, or conference abstracts; (3) case reports or case series research; (4) studies with irrelevant topics; (5) studies with incomplete data or those HR and 95% CI could not be extracted.

The decision to restrict the temporal range of this meta-analysis to studies published between November 2022 and January 2025 was based on both methodological and scientific considerations. Previous meta-analyses covered available literature up to November 2022 [16,17,18,19,20,21,22]. A detailed rationale for this temporal delimitation is provided in Appendix A. Rather than diluting newly published evidence within a broader historical dataset, our aim was to perform a focused, contemporary synthesis to determine whether recent findings confirm, refine, or challenge the previously observed association between PPI use and worse survival outcomes in patients receiving ICIs. Notably, the interaction between PPI and ICI efficacy has become a highly dynamic and actively evolving area of research, spurred by increasing awareness of the gut microbiome’s role in modulating immune responses and therapeutic outcomes. The volume of studies published in the short period following the previous meta-analysis (seven eligible studies in just over two years) reflects this ongoing scientific momentum. A standalone synthesis of these recent studies allows for the detection of temporal trends, potential shifts in clinical practice, and incorporation of methodological advancements (e.g., refined multivariate adjustments, prospective designs, biomarker stratification) that may not have been captured in earlier work. Additionally, recent studies are more likely to reflect current treatment standards, including the use of newer ICIs, combination regimens, and evolving supportive care protocols. As such, their inclusion provides greater relevance to contemporary clinical decision-making.

### 2.2. Data Extraction

The author, year of publication, type of study, sample size, median age, gender, cancer type, immunotherapy strategy (including treatment option and treatment line of ICIs), type of PPIs, number of PPIs users and non-users, timing for PPIs intake, and prognosis information were extracted from each included study.

### 2.3. Primary Objective

This systematic review and meta-analysis aim to assess the impact of PPIs use on time-to-event outcomes, specifically OS and PFS, in cancer patients treated with ICIs.

### 2.4. Quality Assessment

Two independent reviewer teams critically evaluated the methodological quality and risk of bias of the included studies. Quality appraisal was performed using the Methodological Index for Non-Randomized Studies (MINORS) [23] and the Newcastle–Ottawa Scale (NOS) [24]. Appendix A provides a comprehensive description of the methodological instruments employed in this study. Following their assessments, discrepancies between the two teams were systematically reviewed and resolved through consensus discussions.

### 2.5. Statistical Methods

To explore the possible relationship between PPIs, use and survival outcomes—specifically PFS and OS—in patients undergoing ICI therapy, we conducted a comprehensive meta-analysis. Both fixed-effect and random-effects modeling approaches were employed for data synthesis following the DerSimonian and Laird methodology [25]. The fixed-effect model assumes a single true effect across studies, with observed variations attributed entirely to sampling error. In contrast, the random-effects model incorporates between-study variability, postulating that true effect sizes may vary across studies and are drawn from a distribution rather than being identical. This approach yields a combined estimate that represents a weighted average of study-level results, often producing broader confidence intervals and more conservative hazard ratio (HR) estimates, especially in the presence of heterogeneity. The synthesized HRs and their 95% confidence intervals (CIs) are graphically represented via Forest plots. Each plot displays the effect size from individual studies, with the pooled HR shown at the bottom to provide a comprehensive visual summary. For interpretative consistency, HR values were contextualized: an HR of 1.0 reflects no differential risk between PPI users and non-users, while HRs above 1.0 suggest an elevated risk of disease progression or mortality associated with PPI exposure. Where appropriate, HRs were recalculated using Altman’s method [26] to ensure standardization.

Inter-study heterogeneity was evaluated using the I^2^ index [27], which quantifies the proportion of total variation in effect estimates attributable to true heterogeneity rather than random chance. This metric is calculated as I^2^ = 100% × (Q − df)/Q, where Q is Cochran’s heterogeneity statistic and df represents degrees of freedom. Negative I^2^ values were adjusted to zero, and the interpretation follows standard thresholds: <25% (low), 25–50% (moderate), 50–75% (substantial), and >75% (high heterogeneity), with increasing values indicating greater inconsistency that could affect the reliability of pooled results. Potential publication bias was assessed by visual inspection of funnel plots, which map study effect sizes against their standard errors [28]. Symmetrical distribution suggests minimal bias, while asymmetry may indicate selective reporting or small-study effects. Funnel plot asymmetry was formally tested using Egger’s regression and Begg’s rank correlation tests to detect significant deviations.

All analyses were performed using MedCalc Statistical Software version 19.6 (MedCalc Software Ltd., Ostend, Belgium) and Microsoft Excel^®^ for Windows version 2302 (Microsoft Corp., Redmond, WA, USA).

## 3. Results

### 3.1. Baseline Characteristics of Selected Studies

The intent of this analysis is to evaluate whether the use of PPIs influences time-related outcomes, such as PFS and/or OS, in the context of ICI therapy. A total of seven studies were included, comprising a combined population of 10,547 patients with various solid tumors. The PRISMA flowchart illustrates the study selection process (Figure 1), and Table 1 summarizes their key characteristics.

The studies varied in terms of cancer types, ICI regimens, treatment lines, combination with chemotherapy, and reported time-to-outcome endpoints. NSCLC was the most frequently represented tumor type, being the focus in four out of seven studies. Other cancer types included malignant melanoma, urothelial carcinoma, hepatocellular carcinoma, breast cancer, esophageal cancer, kidney cancer, head and neck cancer, and squamous cell skin cancer. The sample size across studies ranged from 127 to 8870. All studies employed retrospective designs and evaluated various ICI agents, including pembrolizumab, nivolumab, atezolizumab, ipilimumab, cemiplimab, avelumab, durvalumab, camrelizumab, simlizumab, and tislelizumab. The line of ICI administration also varied: three studies focused on first-line therapy, while four included ICI use across multiple treatment lines. Regarding concurrent chemotherapy, four studies reported patient cohorts in which ICIs were administered in combination with chemotherapy. The remaining three studies assessed ICI effects without chemotherapy co-administration. In terms of reported outcomes, five studies provided data on both PFS and OS, while two studies reported only OS. This variability in endpoints reflects differences in study design, population, and clinical focus. The methodological quality of the included studies was evaluated using two validated instruments: the MINORS score and the NOS score. As reported in Table 2, all seven studies included in the meta-analysis were retrospective in design but demonstrated overall good methodological quality. MINORS scores ranged from 11 to 14, with none of the studies scoring below 10. Similarly, the NOS scores, which assess quality across three domains—selection, comparability, and outcome—ranged from 6 to 8. No study scored below 6, further supporting the overall robustness of the included evidence, minimizing the risk of bias, and reinforcing the reliability of the subsequent pooled estimates.

### 3.2. Temporal, Quantitative, and Qualitative Characteristics of PPIs Use

Table 3 summarizes the temporal window, frequency, and types of PPIs used in the studies included in the meta-analysis evaluating the impact of PPI exposure on clinical outcomes in patients receiving ICIs. The temporal window of PPI exposure varied across studies. In three studies, PPI use was assessed within a 30-day window prior to the initiation of ICI therapy, while one extended the window to include both 30 days before and after ICI initiation. Two studies considered PPI exposure at the time of ICI treatment, and one evaluated PPI use throughout the ICI treatment course. One study did not specify the timing of PPI use. The proportion of patients exposed to PPIs also varied significantly. The largest cohort was reported by Hong et al., with 2529 patients receiving PPIs and 6341 not exposed. Regarding the qualitative characteristics of PPI use, only three studies specified the types of PPIs administered. One study reported a range of agents, including lansoprazole, rabeprazole, esomeprazole, and vonoprazan, along with the H2-receptor antagonist famotidine. Wang et al. documented the use of omeprazole, pantoprazole, rabeprazole, and additional unspecified agents. Lasagna et al. mentioned pantoprazole as the main agent used, along with other unspecified PPIs. In the remaining four studies, the specific PPI agents were not disclosed.

### 3.3. Impact of PPIs Use on PFS and OS

A meta-analysis was conducted to estimate the pooled effect of PPIs use on PFS and OS in patients with solid tumors, providing a more robust estimate of HRs beyond the fragmented data currently available in the literature.

To assess the robustness and consistency of the PFS-related findings, heterogeneity and potential publication bias were evaluated (Figure 2a). Cochran’s Q test was non-significant, and the I^2^ statistic was 38.09% (95% CI: 0.00–75.38%), indicating moderate heterogeneity among the studies. Funnel plot inspection did not reveal substantial asymmetry, suggesting a low risk of publication bias. For PFS, the pooled analysis included 1367 patients. The fixed-effects model yielded a pooled HR of 1.10 (95% CI: 0.94–1.26), while the random-effects model produced an HR of 1.12 (95% CI: 0.90–1.34) (Figure 2b). Although both estimates suggest a detrimental effect of PPI exposure on PFS, the random-effects model is more appropriate in this context due to the moderate degree of heterogeneity and relatively limited sample size. Based on the random-effects HR of 1.12, PPI use was associated with a 12% increase in the risk of disease progression.

For OS, the pooled analysis included 10,420 patients. Excellent agreement among studies and minimal heterogeneity were documented (Cochran’s Q = non-significant; I^2^ = 0.0%, 95% CI: 0.00–55.49%) (Figure 3a). Funnel plot symmetry and quantitative tests (Egger’s and Begg’s tests) revealed no evidence of publication bias. The pooled HR was 1.18 (95% CI: 1.11–1.25) in both the fixed- and random-effects models (Figure 3b). These findings suggest a consistent and statistically significant adverse impact of PPI use on overall survival. Based on this HR, PPI exposure was associated with an 18% increase in the risk of death in patients with solid tumors.

## 4. Discussion

The results of this meta-analysis provide compelling evidence that the use of PPIs is significantly associated with worsened PFS and OS in patients receiving ICIs for the treatment of solid tumors. PPI use was associated with a 12% increase in the risk of disease progression and a 18% increase in the risk of death. These findings suggest that the influence of PPIs on clinical outcomes, particularly PFS and OS, is an important issue that warrants closer scrutiny. PPIs, including agents such as omeprazole, pantoprazole, and lansoprazole, are widely prescribed to reduce gastric acid secretion. While effective in treating acid-related disorders, their use is associated with disturbances in the gut microbiota [36]. The gastrointestinal tract harbors a diverse array of microbial communities, collectively known as the microbiota, which play a crucial role in modulating various aspects of human health, including immune function [37]. PPIs alter the pH of the stomach, which has downstream effects on the composition and diversity of the gut microbiota [38,39]. A less acidic environment favors the overgrowth of pathogenic bacteria while reducing the abundance of beneficial microbial populations. Research has shown that PPI use is associated with a shift towards dysbiosis, characterized by decreased diversity and the proliferation of harmful microbes such as *Clostridium difficile* and *Enterobacteriaceae* [40,41]. These changes play a crucial role in modulating the immune system. Specifically, one consequence of these alterations is an increase in certain microbial products, such as short-chain fatty acids (SCFAs) produced by the fermentation of dietary fibers, which are involved in regulating immune responses. SCFAs, for example, have been shown to promote the differentiation of regulatory T cells (Tregs) [42]. Tregs are immune-suppressive cells that play a key role in preventing excessive immune activation. Disruptions in the microbiota caused by PPIs use may upset this delicate balance, leading to a suboptimal immune response. In the context of cancer immunotherapy, the microbiota has emerged as an important modulator of response to ICIs. A growing body of evidence suggests that the composition of the gut microbiota can influence the effectiveness of ICIs [43]. For instance, specific microbial species have been linked to enhanced anti-tumor immune responses and improved clinical outcomes in patients receiving ICIs [44].

On the other hand, dysbiosis induced by factors such as PPIs use may impair the ability of the immune system to recognize and attack cancer cells. However, the influence of the microbiota extends beyond immunotherapy. Increasing evidence suggests that the gut microbiota also plays a role in modulating responses to chemotherapy and other biologic agents [45]. For example, certain gut bacteria have been shown to affect the metabolism of chemotherapeutic drugs, influencing their efficacy and toxicity [46]. In preclinical models, the presence of specific bacterial species has been associated with altered drug metabolism and the modulation of chemotherapy-induced side effects.

Additionally, studies in humans have demonstrated that the microbiota can influence the pharmacokinetics of chemotherapeutic agents, potentially impacting treatment outcomes [47]. While the mechanisms underlying these interactions remain incompletely understood, it is evident that the microbiota can modulate the systemic immune response to cancer therapies. Thus, it is plausible that the negative effects of PPIs on microbiota composition may not only impair responses to ICIs but also reduce the efficacy of other cancer treatments, including chemotherapy and targeted biologic therapies. This may explain why the impact of PPIs use on OS, which reflects the cumulative effect of all treatments over time, could be more pronounced than its effect on PFS. OS captures the long-term outcomes of cancer treatment and the patient’s overall health status, including immune function, nutritional status, and comorbidities.

The effects of PPI-induced microbiota dysbiosis may accumulate over time, progressively influencing immune regulation and treatment efficacy as patients undergo multiple lines of therapy. Therefore, while the gut microbiota may play a key role in modulating early responses to immunotherapy, the cumulative effects on OS might be more sensitive to the prolonged and ongoing influence of microbiota dysbiosis on immune function, treatment response, and general health. Moreover, OS is less susceptible to the specific timing of therapy and better reflects the long-term success or failure of the entire treatment regimen, encompassing chemotherapy, ICIs, and other biologic agents.

Despite the robust sample size and the significant findings regarding the impact of PPIs use on PFS and OS, several limitations must be considered. First, the studies included in the meta-analysis were retrospective in design, which inherently carries the risk of bias and confounding factors. The lack of randomization in the allocation of patients to PPIs and non-PPIs groups means that these studies are subject to potential confounders, such as patient comorbidities, co-medications, and other treatment variables, which could affect the observed outcomes. Moreover, the heterogeneity across the studies in terms of tumor types, ICI regimens, and the timing and duration of PPIs exposure presents a challenge in interpreting the pooled results. The variability in the use of different PPI agents (e.g., omeprazole, pantoprazole, lansoprazole, etc.) and the lack of detailed data on clinical variables such as age, gender, and performance status further complicates the interpretation of the findings.

Despite these limitations, the large sample size, the consistency of the findings, and the mechanistic plausibility of PPI-induced microbiota dysbiosis provide strong evidence for the potential role of PPIs in modulating the efficacy of ICIs and, by extension, cancer treatment outcomes. Future studies, ideally prospective and designed with rigorous controls, are needed to further explore the mechanistic underpinnings of these effects. Additionally, studies that include detailed information on microbiota composition and immune response markers would help to clarify the relationship between PPIs use, microbiota modulation, and treatment efficacy.

## 5. Conclusions

In conclusion, the results of this meta-analysis corroborate prior evidence [16] and highlight the potential influence of PPI use on clinical outcomes in patients receiving ICIs for solid malignancies. The negative association with PFS and OS suggests that the use of PPIs may impair the effectiveness of ICIs by altering the gut microbiota and disrupting immune function. While the observed effects are not overwhelming, they are statistically significant and should be taken into consideration when managing cancer patients on immunotherapy. Given the potential implications for patient outcomes, further investigation into the role of PPIs in cancer treatment and their effects on the microbiota is warranted, and clinicians should be mindful of these considerations when prescribing PPIs in the context of immunotherapy.

## Figures and Tables

**Figure 1 cancers-17-02228-f001:**
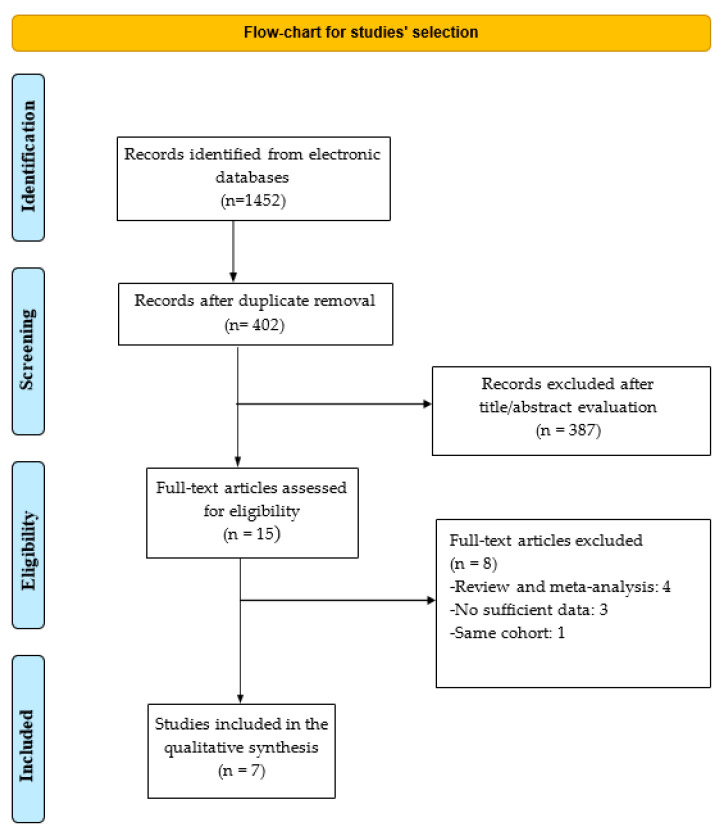
Flowchart of the study selection process.

**Figure 2 cancers-17-02228-f002:**
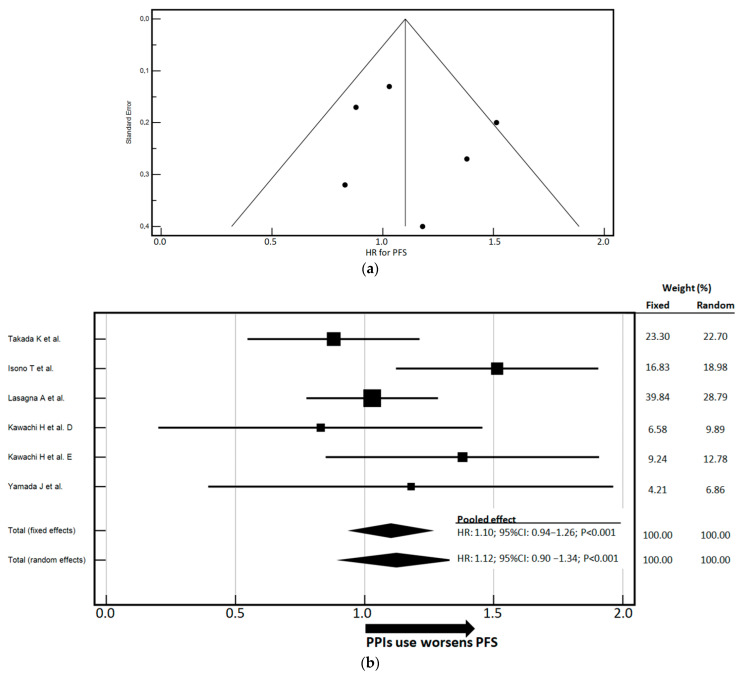
(**a**) The funnel plot displays the standard error on the *Y*-axis and the HR for OS on the *X*-axis. The test for heterogeneity yielded a non-significant Cochran’s Q statistic (Q = 5.05, degrees of freedom = 7, *p* = 0.6545), indicating no substantial between-study variability. The I^2^ statistic, which quantifies the proportion of variance attributable to heterogeneity rather than chance, was 0.0%, with a 95% CI of 0.00% to 55.49%, suggesting negligible inconsistency among the included studies. Evaluation of publication bias showed no significant asymmetry in the funnel plot. Egger’s test yielded an intercept of 0.1371 (95% CI: –1.0092 to 1.2835, *p* = 0.7796), and Begg’s test based on Kendall’s tau also indicated no evidence of bias (τ = 0.03637, *p* = 0.8997). Overall, the results suggest a low risk of publication bias and a high degree of consistency across studies reporting HRs for OS. (**b**) The forest plot illustrates the pooled estimates of the HR for PFS, stratified by PPI use [29,30,32,34,35]. The graphs present these pooled estimates using fixed- and/or random-effects models. To aid interpretation, an arrow on the *x*-axis indicates the trend of PPI use and its association with varying risk, relative to the unit. Weights, which are based on sample size and estimate precision, reflect the relative contribution of each study to the meta-analysis and are displayed to the right of the graphs. The study by Kawachi H et al. is split into two (D, E) because the authors present separate HRs (with corresponding CIs) for two treatment groups (ICIs, ICIs + chemotherapy). CI—Confidence Interval; HR—Hazard Ratio; ICI—Immune Checkpoint Inhibitor; PFS—Progression-Free Survival; PPIs—Proton Pump Inhibitors.

**Figure 3 cancers-17-02228-f003:**
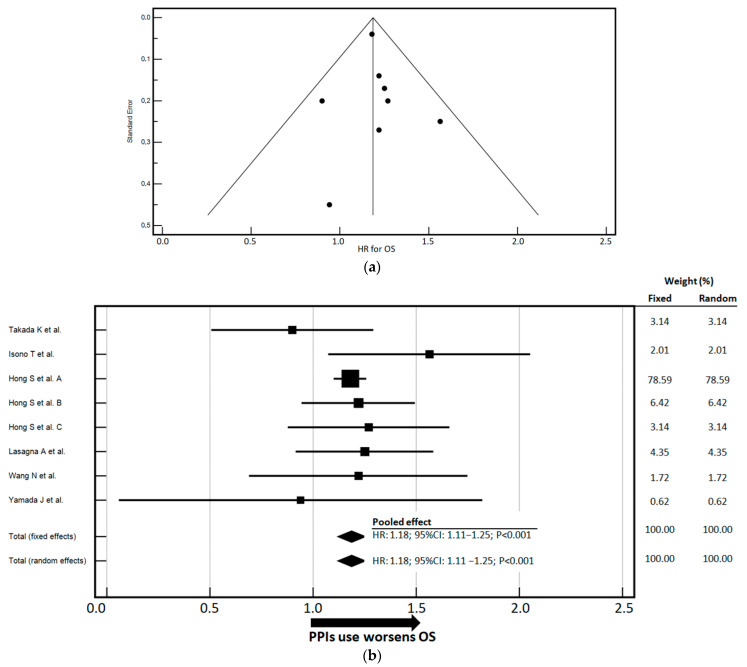
(**a**) The funnel plot displays the standard error on the *Y*-axis and the HR for OS on the *X*-axis. The test for heterogeneity yielded a non-significant Cochran’s Q statistic (Q = 5.05, degrees of freedom = 7, *p* = 0.6545), indicating no substantial between-study variability. The I^2^ statistic, which quantifies the proportion of variance attributable to heterogeneity rather than chance, was 0.0%, with a 95% CI of 0.00% to 55.49%, suggesting negligible inconsistency among the included studies. Evaluation of publication bias showed no significant asymmetry in the funnel plot. Egger’s test yielded an intercept of 0.1371 (95% CI: –1.0092 to 1.2835, *p* = 0.7796), and Begg’s test based on Kendall’s tau also indicated no evidence of bias (τ = 0.03637, *p* = 0.8997). Overall, the results suggest a low risk of publication bias and a high degree of consistency across studies reporting HRs for OS. (**b**) The forest plot displays the aggregated HR estimates for OS, with stratification by the use of PPIs [29,30,31,32,33,35]. These estimates are derived using both fixed- and random-effects models. An arrow on the *x*-axis highlights the direction of PPI usage and its corresponding impact on risk relative to the baseline unit. The weights, shown to the right of the graphs, represent the contribution of each individual study to the overall meta-analysis, adjusted for sample size and the precision of the estimates. The study by Hong S et al. is divided into three sections (A, B, C) as the authors present separate HRs with corresponding CIs for each condition (NSCLC, urothelial carcinoma, malignant melanoma). CI—Confidence Interval; HR—Hazard Ratio; PFS—Progression-Free Survival; PPIs—Proton Pump Inhibitors.

**Table 1 cancers-17-02228-t001:** Characteristics of the selected studies.

Author	Study Design	Patients’ Number	Cancer Type	ICI Treatment	Treatment Line	Included in Association with Chemotherapy	Time-to-Outcome
Takada K et al. [29]	R	198	SCLC	Atezolizumab	First-line	No	OS, PFS
Isono T et al. [30]	R	381	NSCLC	Pembrolizumab, nivolumab, atezolizumab, nivolumab + ipilimumab, nivolumab + ipilimumab	All lines	Yes	OS, PFS
Hong S et al. [31]	R	8870	MM, NSCLC, UC	Pembrolizumab, nivolumab, atezolizumab	All lines	No	OS
Lasagna A et al. [32]	R	363	BC, EC, KC, HNC, MM, NSCLC, SCSC, UC	Nivolumab, pembrolizumab, cemiplimab, atezolizumab, avelumab, durvalumab	All lines	Yes	OS, PFS
Wang N et al. [33]	R	183	HC	Camrelizumab, simlizumab, tislizumab	All lines	Yes	OS
Kawachi H et al. [34]	R	425	NSCLC	Pembrolizumab, atezolizumab	First line	Yes	OS, PFS
Yamada J et al. [35]	R	127	NSCLC	Pembrolizumab, nivolumab, ipilimumab, atezolizumab	Up to third line	Yes	PFS

BC—breast cancer; CT—chemotherapy; EC—esophageal cancer; HC—hepatocellular carcinoma; HNC—head and neck cancer; KC—kidney cancer; MM—malignant melanoma; NSCLC—non-small cell lung cancer; OS—overall survival; PFS—progression-free survival; R—retrospective; SCLC—small cell lung cancer; SCSC—squamous cell skin cancer; UC—urothelial carcinoma.

**Table 2 cancers-17-02228-t002:** Study quality assessment.

First Author, Year	Study Design	MINORS Score	NOS Score
Isono, 2024 [30]	R	12	6
Hong, 2024 [31]	R	11	7
Lasagna, 2023 [32]	R	12	8
Takada, 2024 [29]	R	12	8
Wang, 2024 [33]	R	14	8
Kawachi, 2023 [34]	R	12	8
Yamada, 2024 [35]	R	12	8

MINORS—Methodological Index for Non-Randomized Studies; NOS—Newcastle–Ottawa Scale; R—Retrospective.

**Table 3 cancers-17-02228-t003:** Temporal, quantitative, and qualitative characteristics of the PPIs used.

Author	PPIs Use Window	No. of Patients Treated with PPIs		Type of PPI
		Yes	No	
Takada K et al. [29]	Within 30 days prior to ICI treatment	43	155	NS
Isono T et al. [30]	NS	168	213	Lansoprazole, rabeprazole, esomeprazole, vonoprazan, famotidine
Hong S et al. [31]	Within 30 days before ICI treatment	2529	6341	NS
Lasagna A et al. [32]	Concomitant	189	174	Pantoprazole plus others, NS
Wang N et al. [33]	Within 30 days before and after ICI treatment	88	95	Omeprazole, pantoprazole, rabeprazole, plus others, NS
Kawachi H et al. [34]	Concomitant	134	291	NS
Yamada J et al. [35]	Concomitant	39	88	NS

ICI—immune checkpoint inhibitor; NS—not specified; PPI—proton pump inhibitor.

## Data Availability

No new data were created or analyzed in this study. Data sharing is not applicable to this article.

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
