# Peer review of "Impact of Proton Pump Inhibitor Use on Progression-Free and Overall Survival in Cancer Patients Undergoing Immune Checkpoint Inhibitor Therapy: A Systematic Review and Meta-Analysis of Recent Studies"

_cancers, 2025, doi:10.3390/cancers17132228_

Round 1
Reviewer 1 Report
Comments and Suggestions for Authors
In this manuscript, the authors compiled a pool of over 400 screened articles between November 2022 and January 2025, and applied both fixed-effects and random-effects models to analyze data from 7 selected studies. They found that the use of PPIs was associated with a 12% increased risk of disease progression (HR = 1.12; 95% CI: 0.90–1.34) and an 18% increased risk of mortality (HR = 1.18; 95% CI: 1.11–1.25). The manuscript is supported by extensive clinical data and presents findings of scientific relevance. I would be willing to recommend publication after the following minor revisions are addressed:
-
In lines 250–251, please clarify the distinction between “at the time of ICI treatment” and “throughout the ICI treatment course.”
-
In Table 1, please explain the meaning of the abbreviation “R” under study design.
-
Given the availability of sufficient data, it is recommended that the authors perform a quality assessment specifically on patients with non-small cell lung cancer.
Author Response
Reviewer: In lines 250–251, please clarify the distinction between “at the time of ICI treatment” and “throughout the ICI treatment course.”
Authors: We are grateful to the Reviewer for this comment, which has allowed us to improve the clarity of the text. Indeed, the terminology originally used may have been ambiguous. To avoid confusion and enhance consistency with the terminology adopted across included studies, we have now replaced both expressions with the term “concomitant”, which denotes administration during—neither before nor after—the course of immune checkpoint inhibitor (ICI) therapy.
Reviewer: In Table 1, please explain the meaning of the abbreviation “R” under study design.
Authors: We thank the Reviewer for this observation. The abbreviation “R” stands for “retrospective”, as also indicated in the legend below the table. To ensure greater clarity, we have now made this explicit within the table and its caption.
Reviewer: Given the availability of sufficient data, it is recommended that the authors perform a quality assessment specifically on patients with non-small cell lung cancer.
Authors: We thank the Reviewer for this scientifically relevant suggestion. However, we respectfully believe that a separate quality assessment restricted to NSCLC cases would not be informative in the present context. Several factors limit such an approach. For instance, the heterogeneity introduced by the study by Kawachi et al. (characterised by a wide confidence interval and divergent hazard ratio) would strongly bias the interpretation of NSCLC-specific results. Furthermore, without access to individual patient-level data, key adjustments for treatment timing, tumour burden, and PPI indication are not feasible. For these reasons, and to preserve the methodological coherence of the meta-analysis, we have opted not to conduct a subgroup quality assessment solely for NSCLC.
Reviewer 2 Report
Comments and Suggestions for Authors
Dear Authors,
Systematic review describes well that potential detrimental effect of PPIs on ICI efficacy, highlighting the need for critical assessment of supportive medications during cancer immunotherapy.
Among the potential underlying mechanisms, growing evidence points to the role of the gut microbiota, which can be altered by PPIs, as a plausible contributor to immune modulation and therapeutic response.
The following steps should provide more clear information for readers to enjoy it
Minor revisions are required
iThenticate Report
1) Manuscript matches 29% with other sources – Please minimise similarities.
Reference section
1) Please add up-to-date references in the introduction and discussion section.
Methods section
1) Authors are described very well.
Results section
1) Results are presented in the manuscript in a good way.
Author Response
Reviewer: iThenticate Report
1) Manuscript matches 29% with other sources – Please minimise similarities.
Authors: We acknowledge this point and have taken steps to reduce text similarity. As our group has extensive experience in conducting systematic reviews and meta-analyses, portions of the methodological description (Statistical methods) may reflect standardised phrasing used across similar projects. Nonetheless, we have revised this section to minimise overlap while preserving scientific accuracy.
Reviewer: Reference section
1) Please add up-to-date references in the introduction and discussion section.
Authors: We thank the Reviewer for this suggestion. Indeed, references 3 and 4 were over 10 years old and could be considered outdated. These have now been replaced with more recent and comprehensive studies to ensure alignment with current literature and scientific discourse.
Reviewer: Methods section
1) Authors are described very well.
Authors: We are grateful for the Reviewer’s appreciation of our methodological description.
Reviewer: Results section
1) Results are presented in the manuscript in a good way.
Authors: We sincerely thank the Reviewer. Given the inherent complexity of this type of work, clarity in data presentation is always a major challenge. We are pleased that our efforts to ensure consistency and readability were well received.
Reviewer 3 Report
Comments and Suggestions for Authors
This study explores the effect of proton pump inhibitors combined with immune checkpoint inhibitors on the survival outcomes of cancer patients, which has important clinical significance. However, several similar studies have been published in recent years, which makes this study lack of novelty and significance. For example, PMID 35326555, PMID 35629263, PMID 35860836, PMID 35280736, PMID 36776847, PMID 36872516, Journal of Biological Regulators and Homeostatic Agents 2025, Vol. 39 Issue (1) : 1-6. DOI: 10.23812/j.biol.regul.homeost.agents.20253901.1.
- The included studies were heterogeneous in terms of tumor type, ICIs regimen, PPIs use time, PPIs type, etc., which posed challenges to the interpretation of the results. Not all studies clearly stated the type of PPIs used, which may affect the analysis of the mechanism of PPIs' effects.
- Although this study explored the potential mechanisms by which PPIs affect the efficacy of ICIs, there is a lack of detailed data on the composition of the gut microbiota and immune response markers, which limits a deeper understanding of the mechanisms of PPIs' effects.
Author Response
Reviewer: This study explores the effect of proton pump inhibitors combined with immune checkpoint inhibitors on the survival outcomes of cancer patients, which has important clinical significance. However, several similar studies have been published in recent years, which makes this study lack of novelty and significance. For example, PMID 35326555, PMID 35629263, PMID 35860836, PMID 35280736, PMID 36776847, PMID 36872516, Journal of Biological Regulators and Homeostatic Agents 2025, Vol. 39 Issue (1) : 1-6. DOI: 10.23812/j.biol.regul.homeost.agents.20253901.1.
Authors: This is a phenomenon frequently observed in the literature for this type of analysis. Although international databases are available, multiple research groups often work independently and concurrently on the same topic. However, the studies cited by the Reviewer (PMID: 35326555, 35629263, 35860836, 35280736, 36776847, 36872516, and the article in Journal of Biological Regulators and Homeostatic Agents) completed data collection between July 2021 and November 2022:
PMID 35326555: up to January 2022,
PMID 35629263: up to December 2021,
PMID 35860836: up to March 2022,
PMID 35280736: up to July 2021,
PMID 36776847: up to November 2022,
PMID 36872516: up to September 2021,
J Biol Regul Homeost Agents (2025, Vol. 39, Issue 1): up to September 2022
In contrast, our meta-analysis includes studies published between November 2022 and January 2025, offering a more recent and comprehensive synthesis of available evidence. Moreover, we have described our time-based selection methodology in the Supplementary File, referring to it as a “contiguity-based meta-analytical approach.” This explicitly excludes the temporal scope of the previously mentioned studies. We believe that this temporal extension significantly enhances both the originality and the scientific value of our work.
Reviewer: The included studies were heterogeneous in terms of tumor type, ICIs regimen, PPIs use time, PPIs type, etc., which posed challenges to the interpretation of the results. Not all studies clearly stated the type of PPIs used, which may affect the analysis of the mechanism of PPIs' effects.
Authors: We thank the Reviewer for highlighting these limitations. These sources of heterogeneity are explicitly discussed in the manuscript, and we agree they present interpretative challenges. However, our aim was to synthesise currently available evidence, acknowledging these constraints. We trust that informed readers will be able to interpret our results within the appropriate clinical and methodological context.
Reviewer: Although this study explored the potential mechanisms by which PPIs affect the efficacy of ICIs, there is a lack of detailed data on the composition of the gut microbiota and immune response markers, which limits a deeper understanding of the mechanisms of PPIs' effects.
Authors: We appreciate the Reviewer’s insightful remark. The role of the gut microbiome and immune-related biomarkers in modulating the efficacy of ICIs is indeed compelling. However, as clearly stated in the manuscript, this meta-analysis was not designed to generate mechanistic hypotheses. Rather, our goal was to synthesise clinical outcome data. Mechanistic exploration—such as the impact of PPIs on microbial composition or host immunity—requires dedicated translational or prospective cohort studies. We have cited key publications addressing these issues but believe that detailed mechanistic content exceeds the scope of the current work. We respectfully request that these important topics remain part of the scientific dialogue between authors and reviewers rather than be incorporated into the manuscript itself.
Round 2
Reviewer 3 Report
Comments and Suggestions for Authors
After reading the author's response, I still think that the main problem with this paper is that the research lacks novelty and significance.
Author Response
Comments for Authors from Academic Editor
Please address reviewers' comments.
Authors
Please everything has been completed as requested. Kindly review our responses, including the final section titled 'General Comments to the Academic Editor and Reviewer.
Comments and Suggestions for Authors from Reviewer
Report 1
This study explores the effect of proton pump inhibitors combined with immune checkpoint inhibitors on the survival outcomes of cancer patients, which has important clinical significance. However, several similar studies have been published in recent years, which makes this study lack of novelty and significance. For example, PMID 35326555, PMID 35629263, PMID 35860836, PMID 35280736, PMID 36776847, PMID 36872516, Journal of Biological Regulators and Homeostatic Agents 2025, Vol. 39 Issue (1) : 1-6. DOI: 10.23812/j.biol.regul.homeost.agents.20253901.1.
Authors
As noted in our previous response to Reviewer, the studies cited by the Reviewer (PMID: 35326555, 35629263, 35860836, 35280736, 36776847, 36872516, and the article in Journal of Biological Regulators and Homeostatic Agents) completed data collection between July 2021 and November 2022: PMID 35326555: up to January 2022; PMID 35629263: up to December 2021 PMID 35860836: up to March 2022; PMID 35280736: up to July 2021; PMID 36776847: up to November 2022; PMID 36872516: up to September 2021; J Biol Regul Homeost Agents (2025, Vol. 39, Issue 1): up to September 2022. Accordingly, studies with data collection ending before November 2022 were not included in our meta-analysis. Rather than integrating newly published data into a broader historical dataset, our goal was to conduct a focused and temporally relevant synthesis to assess whether recent findings confirm, refine, or contradict earlier reports on the association between PPI use and poorer survival in patients treated with ICIs. To this end, our meta-analysis includes studies published between November 2022 and January 2025, offering an up-to-date and comprehensive evaluation of the evidence (recent evidence). We refer to this time-based selection strategy as a “contiguity-based meta-analytical approach.”
The studies cited by the Reviewer have been acknowledged in Supplementary File 2, where we explain the rationale and limitations of our temporal inclusion criteria. Additionally, new references have been incorporated into the main text. Please see the revised version.
Reviewer
The included studies were heterogeneous in terms of tumor type, ICIs regimen, PPIs use time, PPIs type, etc., which posed challenges to the interpretation of the results. Not all studies clearly stated the type of PPIs used, which may affect the analysis of the mechanism of PPIs' effects.
Authors
We thank the Reviewer for highlighting these limitations. These sources of heterogeneity are explicitly discussed in the manuscript, and we agree they present interpretative challenges. However, our aim was to synthesise currently available evidence, acknowledging these constraints. We trust that informed readers will be able to interpret our results within the appropriate clinical and methodological context.
Reviewer
Although this study explored the potential mechanisms by which PPIs affect the efficacy of ICIs, there is a lack of detailed data on the composition of the gut microbiota and immune response markers, which limits a deeper understanding of the mechanisms of PPIs' effects.
Authors
We appreciate the Reviewer’s insightful remark. The role of the gut microbiome and immune-related biomarkers in modulating the efficacy of ICIs is indeed compelling. However, as clearly stated in the manuscript, this meta-analysis was not designed to generate mechanistic hypotheses. Rather, our goal was to synthesise clinical outcome data. Mechanistic exploration (such as the impact of PPIs on microbial composition or host immunity) requires dedicated translational or prospective cohort studies. We have cited key publications addressing these issues but believe that detailed mechanistic content exceeds the scope of the current work. We respectfully request that these important topics remain part of the scientific dialogue between authors and reviewers rather than be incorporated into the manuscript itself.
General comments to Academic Editor and Reviewer
We thank the Academic Editor and Reviewer once again for their thoughtful observations. We respectfully note that all these concerns (namely, the temporal range of the meta-analysis, the heterogeneity of included studies and the limited mechanistic data, particularly regarding gut microbiota and immune markers) have already been explicitly acknowledged and addressed in our initial responses, as well as within the manuscript itself. As previously stated, we agree that these aspects introduce interpretative complexity; however, they do not constitute methodological flaws per se for our meta-analysis. Rather, our study reflects the real-world variability of the recent available clinical data and its results have a confirmatory role on the PPIs effect in a temporal frame (2022-2025) characterized by a significant implementation of ICIs indications. Furthermore, our objective was to provide a comprehensive synthesis of recent evidence, with appropriate caveats, rather than to formulate mechanistic conclusions or impose artificial homogeneity.
We hope that the citation of previous literature (included in the new revised version of the manuscript), our transparent discussion of the study limitations (together with the absence of methodological shortcomings) may encourage the Academic Editor and Reviewer to consider the manuscript favorably for publication.